# Unmasking and Exploiting Hidden Strata for Robust and Inclusive Positive Unlabeled Learning

## Abstract

Positive–Unlabeled (PU) learning aims to train a binary classifier using only labeled positive data and a large set of unlabeled samples. Although effective, the state of the art PU learning methods focus on coarse-grained separation between positive and negative classes. In real-world datasets, however, *hidden stratification* frequently occurs, where the positive class comprises multiple fine-grained subclasses with varying prevalence. Ignoring these latent subclasses biases PU classifiers toward dominant subclasses of the positive class, leading to systematic misclassification of rare subclasses. To address this challenge, we propose a subclass-aware PU learning method that first discovers the hidden subclasses through a fully automatic and adaptive graph-based approach. It then leverages the hidden subclasses to select the potential negative examples from the unlabeled set. Comprehensive experimental results demonstrate that the method consistently outperforms the existing PU learning methods on a range of datasets under various distributional settings of the subclasses. A noteworthy property of the proposed method is that it does not require any input about the number of hidden subclasses, thereby making it remarkably robust. To the best of our knowledge, our approach is the first which addresses the hidden subclass issue in PU learning.

## 1 Introduction

Learning a Positive-Unlabeled (PU) classifier is inherently challenging because the negative class information is completely unavailable during training. Unlike conventional supervised classification, where both positive and negative examples are labeled explicitly, PU learning relies only on labeled positive data and a large set of unlabeled samples, which is a mixture of positive and negative instances. Most existing PU learning methods assume that each class is internally homogeneous, focusing on separating positive and negative samples in a coarse-grained manner. However, this assumption rarely holds true in real-world scenarios.

Real datasets are frequently composed of *fine-grained subclasses* within each coarse class. For example, in medical imaging, a single disease label may encompass multiple subtypes with distinct pathological patterns (Dunnmon et al., 2019). Similarly, in object recognition, a high-level label such as "vehicle" may consist of cars, buses, and trucks, each forming a unique subclass (Krizhevsky & Hinton, 2009). This phenomenon, referred to as **hidden stratification**, occurs when training and evaluation samples belonging to the same class do not come from a uniform distribution, but rather from multiple latent subsets with varying prevalence.

Ignoring these subclass structures in PU learning can have consequences. Since negative class information is entirely absent, the classifier is typically biased toward dominant subclasses of the positive class and may fail to generalize to rare or unseen subclasses. If certain fine-grained positive subclasses are missing in the labeled data but appear in the unlabeled data or at test time, the classifier may incorrectly assign them low positive probabilities, causing systematic misclassification. Such biased decision boundaries undermine the robustness and reliability of PU classifiers, particularly in safety-critical domains where consistent performance across data from all subclasses is crucial.

Therefore, addressing hidden stratification in PU learning is essential by explicitly modeling or uncovering fine-grained subclasses within the positive data. This helps to capture diverse and repre-

sentative samples, safeguard rare subclasses from being ignored, and enhance both interpretability and robustness of the PU classifier. To mitigate the effects of hidden stratification, we propose a novel method, **P**ositive-**U**nlabeled Learning method with **S**elf-**c**orrecting **R**egularized **R**isk and **C**onnected **C**omponents (*PU-ScRR-CC*), which explicitly incorporates subclass structure into the PU learning framework. *PU-ScRR-CC* enforces the concept of connected components that refine subclass discovery and enhance label propagation in the labeled positive data. This approach enables a subclass-aware potential negative selection mechanism that better reflects the underlying data distribution in labeled positive dataset.

The principal observations regarding the performance of *PU-ScRR-CC* are: particularly effective under *skewed positive distributions*, where rare subclasses are sparsely represented in the labeled data and maintain competitive performance in almost all cases, even when subclass prevalence is uniform, ensuring the best performing model compared to the state-of-the-art (SOTA) PU methods.

Finally, by aligning subclass-aware representations with decision boundaries, these approaches produce more robust models that generalizes better to unseen positive data distributions.

## 2 THE PROPOSED *PU-ScRR-CC* METHOD

In this work, we introduce a subclass-aware strategy into the PU learning framework and propose **PU-ScRR-CC**, which systematically analyzes latent subclass structures and integrates them into the training process of the proposed *PU-ScRR-CC* PU classifier. By doing so, we enhance the model's ability to generalize beyond dominant subclasses, mitigate the adverse effects of hidden stratification, and achieve more reliable coarse-grained classification in complex real-world datasets.

### 2.1 PROBLEM SETUP

Let $\mathcal{X}$ be the input space. A binary classifier $f$ maps $\mathcal{X}$ consisting of a set of input feature vectors $(x_i)$ to an output space $\mathcal{Y} = \{0, 1\}$ consisting of the binary class labels $(y_i)$. We consider a labeled positive dataset $\mathcal{X}_{\mathsf{p}}$ consisting of $n$ samples $x_1^{\mathsf{p}}, x_2^{\mathsf{p}}, \ldots, x_n^{\mathsf{p}} \in \mathcal{X}$, each annotated with a coarse (superclass) label $y_i^{\mathsf{p}} = 0, \ \forall i \in \{1, \ldots, n\}$. In addition to these observed labels, each sample $x_i^{\mathsf{p}}$ is associated with an unobserved fine-grained subclass label $z_i^{\mathsf{p}} \in \{1, 2, \ldots, K\}$. Also, there is an unlabeled dataset $\mathcal{X}_{\mathsf{u}}$ consisting of $m$ samples $x_1^{\mathsf{u}}, x_2^{\mathsf{u}}, \ldots, x_m^{\mathsf{u}} \in \mathcal{X}$. The coarse label $y_i^{\mathsf{u}}$ of each $x_i^{\mathsf{u}}$ is unknown, but it must be either 0 or 1. The subclass label $z_i^{\mathsf{u}}$ of each $x_i^{\mathsf{u}}$ is also not available.

Therefore the labeled positive dataset $\mathcal{X}_{\mathsf{p}}$ and the unlabeled dataset $\mathcal{X}_{\mathsf{u}}$ are constructed from the associated distribution $\mathcal{D}_{\mathsf{p}}$ and $\mathcal{D}_{\mathsf{u}}$ respectively and $\mathcal{X} = \mathcal{X}_{\mathsf{p}} \cup \mathcal{X}_{\mathsf{u}}$.

$$\mathcal{X}_{\mathsf{p}} = \{x_i^{\mathsf{p}}\}_{i=1}^n \overset{\text{i.i.d.}}{\sim} \mathcal{D}_{\mathsf{p}}, \quad \mathcal{X}_{\mathsf{u}} = \{x_i^{\mathsf{u}}\}_{i=1}^m \overset{\text{i.i.d.}}{\sim} \mathcal{D}_{\mathsf{u}}$$

Our objective is to assign each example from $\mathcal{X}$ to its correct superclass. Given a function class $\mathcal{F}$, the standard approach is to select a classifier $f^* \in \mathcal{F}$ that maximizes overall accuracy under the data distribution $\mathcal{D}$:

$$f^* = \arg \max_{f \in \mathcal{F}} \ \mathbb{E}_{(x,y) \sim \mathcal{D}} \left[ \mathbb{1}_{(f(x)=y)} \right] \tag{1}$$

In addition to overall classification performance, we focus on worst-case test accuracy, which corresponds to the accuracy measured exclusively on samples belonging to the rarest subclass within $\mathcal{X}_{\mathsf{p}}$. The objective of this analysis is to evaluate the effectiveness of mitigating hidden stratification by explicitly identifying hidden subclasses in PU learning. Specifically, we seek to determine whether such subclass-aware approaches yield tangible benefits in terms of robustness. Furthermore, we examine whether worst-case test accuracy remains stable or exhibits significant fluctuations under skewed and uniformly distributed subclass scenarios.

### 2.2 *PU-ScRR-CC*

The initial stage of the framework is dedicated to discovering the latent organization embedded within the labeled positive dataset $\mathcal{X}_{\mathsf{p}}$. In this step, $\mathcal{X}_{\mathsf{p}}$ is converted into an enriched representation $\widehat{\mathcal{X}_{\mathsf{p}}}$, where every positive instance $x_i^{\mathsf{p}}$ receives an inferred hidden subclass label $z_i^{\mathsf{p}}$. This subclass annotation is essential for revealing fine-scale structures inside the positive class and for mitigating

hidden stratification effects. The subclass labels $z_i^{\mathsf{p}}$ are assigned by constructing a similarity graph among the labeled positive samples and extracting its connected components. This graph-based procedure naturally groups data points that are strongly linked in feature space, yielding coherent subclass partitions without imposing spherical or centroid-based assumptions. By incorporating these connectivity-driven clusters, the framework captures subclass-level diversity prior to PU classifier training, enhancing resilience to intra-class variation and subclass imbalance.

HIDDEN SUBCLASS IDENTIFICATION FROM $\mathcal{X}_{\mathsf{p}}$

Let $\mathcal{X}_{\mathsf{p}} = [x_1^{\mathsf{p}}, x_2^{\mathsf{p}}, \ldots, x_n^{\mathsf{p}}]^\top \in \mathbb{R}^{n \times d}$ be the data matrix of labeled positive class with rows $x_i^{\mathsf{p}} \in \mathbb{R}^d$. Define the cosine similarity between points $x_i^{\mathsf{p}}$ and $x_j^{\mathsf{p}}$ as

$$s_{ij} \;=\; \frac{\langle x_i^{\mathsf{p}}, x_j^{\mathsf{p}} \rangle}{\|x_i^{\mathsf{p}}\|_2 \, \|x_j^{\mathsf{p}}\|_2} \quad \text{for } i \neq j$$

Set $s_{ii} \;=\; -\infty$ enforces exclusion of self-neighbors. Let $\mathcal{S} = (s_{ij})_{i,j=1}^n \in \mathbb{R}^{n \times n}$ denote the resulting similarity matrix.

For each index $i \in \{1, \ldots, n\}$, define its (cosine) 1-nearest neighbor(NN),

$$\pi(i) \;\in\; \arg\max_{j \in \{1,\ldots,n\} \setminus \{i\}} s_{ij},$$

with an arbitrary but fixed tie-breaking rule if the maximizer is not unique.

Define an undirected graph $G = (V, E)$ with vertex set $V = \{1, \ldots, n\}$ and edges given by the *symmetrized* 1-NN relation:

$$\{i, j\} \in E \quad \Longleftrightarrow \quad \big(\pi(i) = j\big) \text{ or } \big(\pi(j) = i\big).$$

Equivalently, the adjacency matrix $A \in \{0, 1\}^{n \times n}$ is

$$A_{ij} \;=\; \mathbb{1}_{\{\pi(i)=j \text{ or } \pi(j)=i\}}, \qquad A_{ii} = 0.$$

Let the connected components of $G$ be $\mathcal{C}_1, \ldots, \mathcal{C}_K$, where $\mathcal{C}_k \subseteq V$, $\mathcal{C}_k \cap \mathcal{C}_{k'} = \emptyset$ for $k \neq k'$, and $\bigcup_{k=1}^K \mathcal{C}_k = V$. Define the cluster-label map $\mathcal{I} : \{1, \ldots, n\} \to \{1, \ldots, K\}$ by

$$\mathcal{I}(i) \;=\; k \quad \text{iff} \quad i \in \mathcal{C}_k.$$

The final clustering assignment $\mathcal{Z} = (z_1^{\mathsf{p}}, \ldots, z_n^{\mathsf{p}})^\top$ yields the subclass-aware labeled set $\widehat{\mathcal{X}_{\mathsf{p}}}$, explicitly capturing fine-grained structure in the positive class. This refined representation is combined with $\mathcal{X}_{\mathsf{u}}$ to form $\mathcal{X}^*$, on which *PU-ScRR* is applied. Incorporating subclass information enables balanced learning across both common and rare subclasses, improving robustness and worst-case performance.

## 2.3 *PU-ScRR*

The proposed *PU-ScRR-CC* builds on the *PU-ScRR* framework to mitigate hidden stratification in the labeled positive set $\mathcal{X}_{\mathsf{p}}$. It clusters $\mathcal{X}_{\mathsf{p}}$ via similarity-based connected-component analysis, assigning inferred subclass labels to obtain $\widehat{\mathcal{X}_{\mathsf{p}}}$. In the warm-start phase, high-confidence negatives from $\mathcal{X}_{\mathsf{u}}$ form the candidate negative set $\widetilde{\mathcal{X}_{\mathsf{n}}}$. A deep classifier $g_{\boldsymbol{\theta}}$ is then trained on $\widehat{\mathcal{X}_{\mathsf{p}}}$ and $\widetilde{\mathcal{X}_{\mathsf{n}}}$ to learn robust decision boundaries under the PU setting. The details of each phase are discussed in the subsequent sections.

WARM-START PHASE: SELECTING POTENTIAL NEGATIVES FROM $\mathcal{X}_{\mathsf{u}}$

The first phase of *PU-ScRR* focuses on gathering reliable information about the negative class by extracting representative samples from the unlabeled dataset. The intuition is to identify data points in $\mathcal{X}_{\mathsf{u}}$ that are least similar to the clustered labeled positive set $\widehat{\mathcal{X}_{\mathsf{p}}}$ and thus most likely to be negative. This step provides a "warm start" for training the PU classifier in the subsequent phase.

It is well-known that neural networks tend to learn "easy" samples—those with clearly separable features—more rapidly than "hard" samples that exhibit subtle or ambiguous characteristics (Chatterjee, 2020). Leveraging this property, we design a mechanism to preferentially identify easy negative examples from the unlabeled data.

---

**Algorithm 1** Potential negative sample selection algorithm

---

**Input:** Clustered labeled positive data $\widehat{\mathcal{X}}_{\mathsf{p}}$, unlabeled data $\mathcal{X}_{\mathsf{u}}$, number of epochs $k$, potential negative sampler model $h_{\bar{\theta}}$. $S_K = \{1, 2, \ldots, K\}$ be the set of all possible hidden subclass labels in $\widehat{\mathcal{X}}_{\mathsf{p}}$

**Output:** Trained potential negative sampler $h_{\bar{\theta}}^*$, potential negative sample set $\widetilde{\mathcal{X}}_{\mathsf{n}}$.

---

1: Initialize $\widetilde{\mathcal{X}}_{\mathsf{n}} \leftarrow \varnothing$.
2:     Treat all unlabeled data as negatives: $\mathcal{X}_{\mathsf{n}} \leftarrow \mathcal{X}_{\mathsf{u}}$.
3:     Initialize model $h_{\bar{\theta}}$ and optimization routine $\mathcal{A}$.
4: **for** $i = 1$ to $k$ **do**
5:     Shuffle $(\mathcal{X}_{\mathsf{p}}, \mathcal{X}_{\mathsf{n}})$ into $B$ mini-batches: $(\mathcal{X}_{\mathsf{p}}^j, \mathcal{X}_{\mathsf{n}}^j)$, $j = 1, \ldots, B$.
6:     **for** $j = 1$ to $B$ **do**
7:         Compute gradient:

$$\nabla_{\bar{\theta}} \left( \frac{1}{|\mathcal{X}_{\mathsf{p}}^j|} \sum_{\substack{x_{\mathsf{p}} \in \mathcal{X}_{\mathsf{p}}^j \\ z_{\mathsf{p}} \in S_K}} \ell(h_{\bar{\theta}}(x_{\mathsf{p}}), z_{\mathsf{p}}) + \frac{1}{|\mathcal{X}_{\mathsf{n}}^j|} \sum_{x_{\mathsf{n}} \in \mathcal{X}_{\mathsf{n}}^j} \ell(h_{\bar{\theta}}(x_{\mathsf{n}}), (C+1)) \right)$$

        and update $\bar{\theta}$ with algorithm $\mathcal{A}$.
8:     **end for**
9: **end for**

---

Let $\widehat{\mathcal{X}}_{\mathsf{p}} = \{x_i^{\mathsf{p}}\}_{i=1}^n$ denote the set of labeled positives associated with a hidden subclass label $z_i^{\mathsf{p}}$, and $\mathcal{X}_{\mathsf{u}} = \{x_i^{\mathsf{u}}\}_{i=1}^m$ denote the set of unlabeled samples. $z_i^{\mathsf{p}} \in \{1, 2, \ldots, K\}$, where $K$ is the total number of possible hidden subclasses in $\mathcal{X}_{\mathsf{p}}$. Initially, we treat $\mathcal{X}_{\mathsf{u}}$ as if it were entirely negative and train a $(K+1)$-ary classifier $h_{\bar{\theta}}$ for $k$ epochs (with $k$ chosen empirically). This short training period emphasizes the easy-to-learn negative samples while limiting the influence of ambiguous ones.

The classifier $h_{\bar{\theta}}$ consists of a feature extractor $\varphi(\cdot)$ for generating representations $\varphi(x)$ and a feed-forward head $\varsigma(\cdot)$ trained with cross-entropy loss. After warm-start training, each unlabeled sample $x_i^{\mathsf{u}}$ is assigned a confidence score $\beta_i = h_{\bar{\theta}}^*(x_i^{\mathsf{u}})$. The top-scoring samples form the candidate negative set $\widetilde{\mathcal{X}}\mathsf{n}$ with $|\widetilde{\mathcal{X}}\mathsf{n}| = |\widehat{\mathcal{X}}_{\mathsf{p}}|$, while low-confidence samples are treated as unreliable. The warm-start procedure is detailed in Algorithm 1.

### TRAINING PHASE: LEARNING PU CLASSIFIER IN SUPERVISED PARADIGM

In the second phase, the PU classifier $g_\theta$ is trained using $\mathcal{X}\mathsf{p}$ as positives and $\widetilde{\mathcal{X}}\mathsf{n}$ as negatives. It reuses the pre-trained feature extractor $\varphi(\cdot)$ from the warm start, feeding representations $v_i = \varphi(x_i)$ into two heads: $\omega_1(\cdot)$ for classification and $\omega_2(\cdot)$ for confidence estimation.

The overall classifier output is given by $g_\theta(x) = (\mu, \kappa)$, $\mu = \omega_1(v)$, $\kappa = \omega_2(v)$, where $\kappa_i$ quantifies the likelihood that an unlabeled sample belongs to the negative class. Formally, the classifier is a mapping $g_\theta \colon \mathcal{X}^* \to [0, 1] \times \mathbb{R}$, $\mathcal{X}^* = \mathcal{X}_{\mathsf{p}} \cup \widetilde{\mathcal{X}}_{\mathsf{n}}$.

To train $g_\theta$, we define empirical risks associated with partially labeled data. For a given sample set $X = \{x_1, \ldots, x_n\}$ and classifier output $\mu_i = \omega_1(\varphi(x_i))$,

$$\widehat{L}_+(g_\theta; X) = \frac{1}{n} \sum_{i=1}^n \ell(\mu_i, 1), \tag{2}$$

$$\widehat{L}_-(g_\theta; X) = \frac{1}{n} \sum_{i=1}^n \kappa_i^+ \, \ell(\mu_i, 0), \tag{3}$$

where $\ell(\mu_i, y_i) = -\big[y_i \ln \mu_i + (1 - y_i) \ln(1 - \mu_i)\big]$ is the binary cross-entropy loss and $\kappa_i^+ = \max(\kappa_i, 0)$ ensures negative confidence scores of an unlabeled data. The first term $\widehat{L}_+$ measures the risk under the assumption that samples are truly positive, while $\widehat{L}_-$ evaluates the risk assuming they are truly negative. Combining these, the objective function for PU classification in *PU-ScRR* is

---

**Algorithm 2** PU classification algorithm (*PU-ScRR*)

---

**Input:** Labeled positive data $\mathcal{X}_{\mathsf{p}}$, potential negative data $\widetilde{\mathcal{X}_{\mathsf{n}}}$, hyperparameter $\delta \in (0,1)$, PU classifier $g_{\boldsymbol{\theta}}$.

**Output:** Trained PU classifier $g_{\boldsymbol{\theta}}^*$.

---

1: Form the combined training set $\mathcal{X}^* \leftarrow \mathcal{X}_{\mathsf{p}} \cup \widetilde{\mathcal{X}_{\mathsf{n}}}$. Initialize the classifier $g_{\boldsymbol{\theta}}$ and a stochastic optimizer $\mathcal{A}$.

2: **while** training error is not converged **do**

3:   Partition the combined dataset $\mathcal{X}^*$ into $B$ mini-batches, denoting the $i$-th mini-batch as $\mathcal{X}_i^* = (\mathcal{X}_{\mathsf{p}}^{(i)}, \widetilde{\mathcal{X}_{\mathsf{n}}}^{(i)})$.

4:   $\widehat{L}_+(g_{\boldsymbol{\theta}}; \mathcal{X}_i^*) \leftarrow \frac{1}{|\mathcal{X}_{\mathsf{p}}^i|} \sum_{x_{\mathsf{p}} \in \mathcal{X}_{\mathsf{p}}^i} \ell(\mu_{\mathsf{p}}, 1), \quad \mu_{\mathsf{p}} = \omega_1(\varphi(x_{\mathsf{p}}))$ for each $x_{\mathsf{p}} \in \mathcal{X}_{\mathsf{p}}^i$

5:   $\widehat{L}_-(g_{\boldsymbol{\theta}}; \mathcal{X}_i^*) \leftarrow \frac{1}{|\widetilde{\mathcal{X}_{\mathsf{n}}^i}|} \sum_{x_{\mathsf{n}} \in \widetilde{\mathcal{X}_{\mathsf{n}}^i}} \kappa_{\mathsf{n}}^+ \ell(\mu_{\mathsf{n}}, 0)$, where $\kappa_{\mathsf{n}}^+ = \max\big(\omega_2(\varphi(x_{\mathsf{n}})), 0\big)$ for each $x_{\mathsf{n}} \in \widetilde{\mathcal{X}_{\mathsf{n}}^i}$.

6:   $\mathsf{Reg}(g_{\boldsymbol{\theta}}; \mathcal{X}_i^*) \leftarrow \delta \sum_{x_i \in \mathcal{X}_i^*} |1 - \kappa_i^+|$

7:   Set the gradient
$$\nabla_\theta \big(\widehat{L}_+(g_{\boldsymbol{\theta}}; \mathcal{X}_i^*) + \widehat{L}_-(g_{\boldsymbol{\theta}}; \mathcal{X}_i^*) + \mathsf{Reg}(g_{\boldsymbol{\theta}}; \mathcal{X}_i^*)\big)$$

8: **end while**

---

defined as:

$$\widehat{R}_{\mathsf{PU\text{-}ScRR}}(\boldsymbol{\theta}; g_{\boldsymbol{\theta}}) = \frac{1}{|\mathcal{X}_{\mathsf{p}}|} \sum_{x_{\mathsf{p}} \in \mathcal{X}_{\mathsf{p}}} \ell(\mu_{\mathsf{p}}, 1) + \frac{1}{|\widetilde{\mathcal{X}_{\mathsf{n}}}|} \sum_{x_{\mathsf{n}} \in \widetilde{\mathcal{X}_{\mathsf{n}}}} \lambda_{\mathsf{n}}^+ \ell(\mu_{\mathsf{n}}, 0) + \delta \sum_{x_i \in \mathcal{X}^*} |1 - \kappa_i^+|, \quad (4)$$

where $\delta$ is a regularization coefficient encouraging confidence scores to remain close to unity for high-confidence negative samples. Algorithm 2 summarizes the training procedure for the PU classifier $g_{\boldsymbol{\theta}}$ based on the above risk formulation.

## 3 EXPERIMENTS

To evaluate the efficiency of the proposed algorithm *PU-ScRR-CC*, we implement the algorithms on the CIFAR-10 (Krizhevsky & Hinton, 2009), CIFAR-100 (Krizhevsky & Hinton, 2009), Fashion-MNIST (Xiao et al., 2017) and STL-10 (Coates et al., 2011) datasets.

### 3.1 DETAILS OF CLASSIFICATION TASKS

We have designed and performed several classification experiments on the CIFAR-10, CIFAR-100, Fashion-MNIST, and STL-10 datasets. We have conducted a total of 8 distinct binary classification tasks depending on the chosen classes in labeled positive dataset.

The binary classification tasks span CIFAR-10, CIFAR-100, Fashion-MNIST, and STL-10, differing mainly in the subclass composition of the labeled positive set ($\mathcal{X}\mathsf{p}$) and unlabeled set ($\mathcal{X}\mathsf{u}$). CIFAR-10-1 uses two animal subclasses in $\mathcal{X}\mathsf{p}$ and one non-animal subclass in $\mathcal{X}\mathsf{u}$, while CIFAR-10-2 swaps all animal and non-animal subclasses between the sets. CIFAR-100 tasks classify aquatic mammals vs. fish, with CIFAR-100-1 using 2 vs. 1 subclasses and CIFAR-100-2 using all 5 subclasses for both classes. Fashion-MNIST tasks classify topwear vs. others, varying only in the number of subclasses per class. STL-10-1 and STL-10-2 define vehicle–animal and animal–vehicle tasks, each with two subclasses in $\mathcal{X}\mathsf{p}$ and one in $\mathcal{X}\mathsf{u}$.

### 3.2 ANALYSIS OF EXPERIMENTAL RESULTS

To comprehensively evaluate the effectiveness of *PU-ScRR-CC*, we conduct eight controlled experiments across four widely used benchmark datasets: CIFAR-10, CIFAR-100, Fashion-MNIST, and STL-10. For comparison, we include several state-of-the-art (SOTA) PU learning approaches.

Table 1: Comparison of mean overall test accuracies (with standard deviations) of *PU-ScRR-CC* against other SOTA methods on CIFAR–10 and CIFAR–100 datasets.

| Dataset | Task Name | $\alpha_p$ | Class label ratio in $\mathcal{X}_p$ | Methods | | | | | | |
| --- | --- | --- | --- | --- | --- | --- | --- | --- | --- | --- |
| | | | | *uPU* | *nnPU* | *(TED)^n* | *HolisticPU* | *LaGAM* | *PU-ScRR* | *PU-ScRR-CC* |
| CIFAR–10 | CIFAR-10-1 | 0.8 | 9:1 | 80.8 ± 1.3 | 74.1 ± 1.7 | 68.1 ± 0.9 | 80.1 ± 0.8 | 79.4 ± 1.5 | 80.5 ± 0.5 | **82.9 ± 0.3** |
| | | | 1:1 | 84.1 ± 0.8 | 83.8 ± 0.7 | 82.4 ± 0.1 | 83.0 ± 2.8 | 84.6 ± 0.9 | **85.8 ± 0.8** | 85.5 ± 0.5 |
| | | | 1:9 | 82.1 ± 0.8 | 83.1 ± 2.2 | 81.3 ± 1.2 | 72.1 ± 6.0 | 80.9 ± 0.7 | 82.4 ± 0.9 | **84.0 ± 0.5** |
| | | 0.5 | 9:1 | 77.4 ± 2.0 | 81.7 ± 0.8 | 78.6 ± 1.7 | 85.8 ± 1.8 | 86.3 ± 0.8 | 86.1 ± 0.4 | **87.2 ± 0.3** |
| | | | 1:1 | 83.7 ± 0.9 | 85.7 ± 0.4 | 88.8 ± 0.6 | 86.5 ± 1.2 | 88.8 ± 0.5 | **89.7 ± 0.6** | 88.5 ± 0.6 |
| | | | 1:9 | 83.0 ± 1.0 | 85.4 ± 0.8 | 87.5 ± 0.6 | 86.6 ± 1.2 | 87.0 ± 1.0 | 87.5 ± 0.7 | **88.4 ± 0.3** |
| | | 0.2 | 9:1 | 77.4 ± 1.8 | 83.4 ± 1.0 | 87.8 ± 0.4 | 86.6 ± 0.2 | 87.2 ± 0.9 | 88.0 ± 0.4 | **88.6 ± 0.6** |
| | | | 1:1 | 83.0 ± 1.4 | 88.3 ± 0.4 | 88.7 ± 0.3 | 87.6 ± 0.8 | 89.9 ± 0.2 | **90.0 ± 0.7** | 89.5 ± 0.3 |
| | | | 1:9 | 78.0 ± 2.2 | 86.0 ± 0.9 | 89.0 ± 0.7 | 86.8 ± 0.8 | 88.2 ± 0.3 | 88.0 ± 0.6 | **89.0 ± 0.2** |
| | CIFAR-10-2 | 0.8 | 8:4:2:1 | 81.6 ± 1.9 | 81.4 ± 1.9 | 76.3 ± 0.8 | 83.0 ± 0.4 | 84.3 ± 0.6 | 84.5 ± 0.8 | **86.5 ± 0.3** |
| | | | 4:2:1:1 | 82.0 ± 1.8 | 83.4 ± 1.3 | 77.8 ± 0.5 | 81.8 ± 0.3 | 84.4 ± 0.4 | 87.6 ± 0.5 | **88.2 ± 0.5** |
| | | 0.5 | 8:4:2:1 | 82.6 ± 1.4 | 83.0 ± 0.7 | 85.3 ± 0.9 | 87.5 ± 0.9 | 87.0 ± 0.5 | 87.4 ± 0.4 | **89.2 ± 0.3** |
| | | | 4:2:1:1 | 81.7 ± 2.2 | 83.3 ± 2.0 | 86.7 ± 1.3 | 85.7 ± 0.3 | 87.2 ± 0.1 | **91.7 ± 0.6** | 89.5 ± 1.0 |
| | | 0.2 | 8:4:2:1 | 88.0 ± 0.3 | 90.2 ± 0.2 | 90.7 ± 0.2 | 89.0 ± 5.7 | 90.8 ± 0.4 | 91.3 ± 0.1 | **91.6 ± 0.6** |
| | | | 4:2:1:1 | 88.9 ± 0.5 | 91.1 ± 0.4 | 91.5 ± 0.3 | 87.7 ± 3.5 | 91.2 ± 0.2 | **93.6 ± 0.9** | 92.5 ± 0.3 |
| CIFAR–100 | CIFAR-100-1 | 0.8 | 9:1 | 59.2 ± 0.9 | 59.3 ± 0.5 | 60.0 ± 0.2 | 55.5 ± 0.8 | 51.1 ± 0.5 | 58.5 ± 0.6 | **65.5 ± 0.1** |
| | | | 1:1 | 63.0 ± 0.5 | 50.0 ± 0.0 | 64.3 ± 0.4 | 61.4 ± 0.8 | 60.0 ± 0.8 | **67.9 ± 0.2** | 61.8 ± 0.6 |
| | | | 1:9 | 63.6 ± 0.7 | 68.9 ± 0.9 | 68.5 ± 0.4 | 63.0 ± 0.6 | 67.2 ± 0.5 | 69.0 ± 0.2 | **69.9 ± 0.7** |
| | | 0.5 | 9:1 | 63.6 ± 0.8 | 67.1 ± 0.4 | 66.3 ± 0.8 | 63.1 ± 0.5 | 56.8 ± 0.7 | 65.0 ± 0.6 | **68.6 ± 0.1** |
| | | | 1:1 | 64.1 ± 0.6 | 69.8 ± 0.1 | 71.4 ± 0.8 | 66.4 ± 0.1 | 70.2 ± 0.6 | **72.3 ± 0.2** | 71.6 ± 0.2 |
| | | | 1:9 | 70.1 ± 0.6 | 70.1 ± 0.4 | 72.3 ± 0.3 | 69.8 ± 0.5 | 70.2 ± 0.4 | 71.3 ± 0.8 | **74.5 ± 0.4** |
| | | 0.2 | 9:1 | 66.9 ± 0.1 | 70.4 ± 0.4 | 71.5 ± 0.4 | 71.3 ± 0.9 | 66.4 ± 0.6 | 67.6 ± 0.9 | **72.4 ± 0.1** |
| | | | 1:1 | 68.4 ± 0.3 | 50.0 ± 0.0 | 72.4 ± 0.5 | 72.5 ± 0.9 | 71.9 ± 0.5 | **73.5 ± 0.1** | 73.5 ± 0.7 |
| | | | 1:9 | 72.6 ± 0.8 | 71.1 ± 0.6 | 72.8 ± 0.5 | 73.9 ± 0.7 | 73.5 ± 0.3 | 74.2 ± 0.2 | **74.8 ± 0.8** |
| | CIFAR-100-2 | 0.8 | 5:4:3:2:1 | 75.4 ± 0.9 | 77.6 ± 0.1 | 77.5 ± 0.3 | 71.0 ± 0.8 | 65.1 ± 0.1 | 76.0 ± 0.7 | **79.0 ± 0.8** |
| | | | 1:1:1:1:1 | 77.0 ± 0.7 | 78.0 ± 0.1 | 78.4 ± 0.5 | 69.6 ± 0.1 | 66.1 ± 0.5 | **81.1 ± 0.6** | 80.1 ± 1.0 |
| | | 0.5 | 5:4:3:2:1 | 74.5 ± 0.2 | 75.8 ± 0.7 | 74.7 ± 0.6 | 65.0 ± 0.2 | 74.0 ± 0.8 | 75.8 ± 0.8 | **76.8 ± 0.5** |
| | | | 1:1:1:1:1 | 69.6 ± 0.9 | 72.8 ± 0.5 | 75.1 ± 0.3 | 75.4 ± 0.8 | 74.5 ± 0.5 | **78.6 ± 0.2** | 77.7 ± 0.1 |
| | | 0.2 | 5:4:3:2:1 | 77.0 ± 0.6 | 72.3 ± 0.4 | 76.7 ± 0.6 | 80.1 ± 0.2 | 77.2 ± 0.5 | 80.3 ± 0.4 | **81.1 ± 0.8** |
| | | | 1:1:1:1:1 | 73.0 ± 0.6 | 71.5 ± 0.6 | 78.6 ± 0.7 | 80.3 ± 0.5 | 77.4 ± 0.6 | **81.9 ± 0.6** | 81.7 ± 0.1 |

Specifically, *uPU* and *nnPU* assume prior knowledge of the positive class prior $\alpha_p$, whereas *(TED)^n* relies on its estimation. *HolisticPU* takes a different route by estimating trend scores for unlabeled samples to identify and resample potential negatives. In contrast, *PU-ScRR-CC* avoids both class prior estimation and heuristic resampling, making it more broadly applicable in real-world scenarios where subclass distributions and priors are unknown.

Table 1 presents the mean overall test accuracy values of *PU-ScRR-CC* and other SOTA PU classifiers with standard deviations on CIFAR–10 and CIFAR–100 datasets. *PU-ScRR-CC* surpasses SOTA methods by an approx margin of $1\% - 7\%$ in almost all cases, regardless of the $\alpha_p$ values and the non-uniform subclass distribution in $\mathcal{X}_p$ on CIFAR–10 and CIFAR–100 datasets. When subclasses are uniformly represented, *PU-ScRR* outperforms *PU-ScRR-CC* by $0\% - 6\%$ approximately. The best performing results are indicated in **bold**.

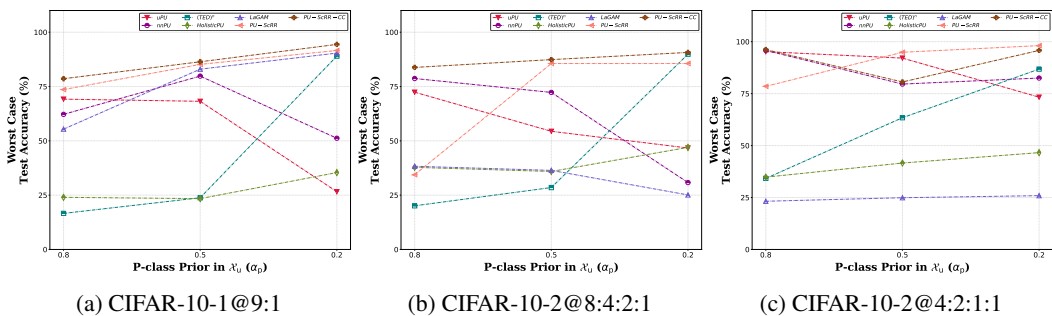

(a) CIFAR-10-1@9:1  (b) CIFAR-10-2@8:4:2:1  (c) CIFAR-10-2@4:2:1:1

Figure 1: Comparison of worst-case test accuracies of *PU-ScRR-CC* against other SOTA methods under varying proportions of positive data in the unlabeled set $\mathcal{X}_u$ on CIFAR–10 dataset.

The worst-case test accuracies of *PU-ScRR-CC* and other SOTA methods on the CIFAR–10 dataset are presented in Figure 1. *PU-ScRR-CC* outperforms almost all the cases with a minimum of $4.4\%$

average guaranteed improved worst-case test accuracy. Figure 1c indicates that *PU-ScRR* surpasses *PU-ScRR-CC* as the top-performing model. It is noteworthy that the ship subclass samples are uniformly represented in $\mathcal{X}_\mathsf{p}$.

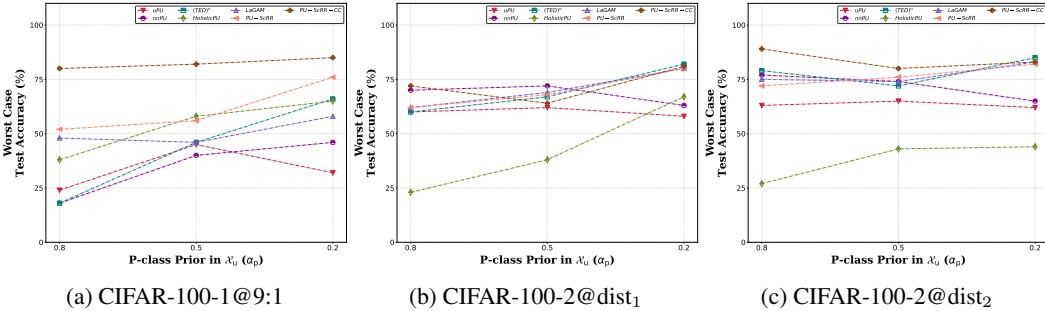

|  |  |  |
|---|---|---|
| (a) CIFAR-100-1@9:1 | (b) CIFAR-100-2@$dist_1$ | (c) CIFAR-100-2@$dist_2$ |

Figure 2: Comparison of worst-case test accuracies of *PU-ScRR-CC* against other SOTA methods under varying proportions of positive data in the unlabeled set $\mathcal{X}_\mathsf{u}$ on CIFAR–100 datasets.

*PU-ScRR-CC* achieves significant improvement in worst-case test accuracy across nearly all scenarios. The highest improvement margin is observed in figure 2a, which is approx. 37% on average across all considered $\alpha_\mathsf{p}$. Figure 2c demonstrates the superior performance of *PU-ScRR-CC*, even when all subclasses are uniformly represented in $\mathcal{X}_\mathsf{p}$.

Table 2 reports the mean test accuracies (with standard deviations) of *PU-ScRR-CC* and other state-of-the-art PU classifiers on Fashion–MNIST and STL–10 datasets. *PU-ScRR-CC* surpasses SOTA methods by approximately 1%–6% across most cases, regardless of $\alpha_\mathsf{p}$ values or subclass imbalance in $\mathcal{X}_\mathsf{p}$, with best results highlighted in **bold**. When subclasses are uniformly represented, *PU-ScRR* outperforms *PU-ScRR-CC* by 0%−2% approximately.

Table 2: Comparison of mean overall test accuracies (with standard deviations) of *PU-ScRR-CC* against other SOTA methods on Fashion–MNIST and STL–10 datasets.

| Dataset | Task Name | $\alpha_\mathsf{p}$ | Class label ratio in $\mathcal{X}_\mathsf{p}$ | Methods | | | | | | |
|---|---|---|---|---|---|---|---|---|---|---|
|  |  |  |  | *uPU* | *nnPU* | $(TED)^n$ | *HolisticPU* | *LaGAM* | *PU-ScRR* | *PU-ScRR-CC* |
| Fashion–MNIST | F-MNIST-1 | 0.8 | 9:1 | $89.2 \pm 2.9$ | $95.5 \pm 0.4$ | $76.8 \pm 2.7$ | $80.3 \pm 1.7$ | $87.1 \pm 3.4$ | $96.6 \pm 0.8$ | $\mathbf{97.4 \pm 1.0}$ |
|  |  |  | 1:1 | $94.5 \pm 0.1$ | $96.5 \pm 0.2$ | $90.1 \pm 1.5$ | $79.1 \pm 2.5$ | $97.7 \pm 0.3$ | $98.5 \pm 0.3$ | $\mathbf{98.6 \pm 0.2}$ |
|  |  |  | 1:9 | $88.3 \pm 5.1$ | $94.7 \pm 1.8$ | $91.6 \pm 2.3$ | $78.6 \pm 3.6$ | $84.7 \pm 1.1$ | $97.8 \pm 0.5$ | $\mathbf{98.4 \pm 0.1}$ |
|  |  | 0.5 | 9:1 | $92.3 \pm 2.9$ | $97.2 \pm 0.1$ | $86.0 \pm 1.0$ | $95.4 \pm 1.2$ | $97.0 \pm 0.2$ | $97.7 \pm 0.1$ | $\mathbf{98.1 \pm 0.3}$ |
|  |  |  | 1:1 | $95.9 \pm 0.2$ | $98.1 \pm 0.2$ | $96.7 \pm 0.3$ | $97.5 \pm 1.5$ | $98.2 \pm 0.5$ | $\mathbf{98.9 \pm 0.2}$ | $98.7 \pm 0.1$ |
|  |  |  | 1:9 | $83.1 \pm 2.6$ | $97.8 \pm 0.3$ | $97.5 \pm 0.3$ | $97.9 \pm 1.0$ | $95.0 \pm 0.7$ | $98.7 \pm 0.2$ | $\mathbf{98.9 \pm 0.2}$ |
|  |  | 0.2 | 9:1 | $87.1 \pm 2.7$ | $96.0 \pm 0.3$ | $96.9 \pm 0.2$ | $95.0 \pm 0.2$ | $96.9 \pm 0.3$ | $98.6 \pm 0.3$ | $\mathbf{98.9 \pm 0.2}$ |
|  |  |  | 1:1 | $90.9 \pm 0.8$ | $97.8 \pm 0.1$ | $97.0 \pm 0.1$ | $96.1 \pm 0.3$ | $97.4 \pm 0.2$ | $\mathbf{98.9 \pm 0.4}$ | $98.3 \pm 0.5$ |
|  |  |  | 1:9 | $83.5 \pm 2.1$ | $96.1 \pm 0.6$ | $96.8 \pm 0.1$ | $96.7 \pm 0.4$ | $95.7 \pm 0.4$ | $98.0 \pm 0.3$ | $\mathbf{98.6 \pm 0.3}$ |
|  | F-MNIST-2 | 0.8 | 8:4:2:1 | $92.1 \pm 0.2$ | $93.3 \pm 0.9$ | $84.1 \pm 2.0$ | $77.7 \pm 2.0$ | $86.1 \pm 1.3$ | $94.1 \pm 0.4$ | $\mathbf{95.7 \pm 0.6}$ |
|  |  |  | 4:2:1:1 | $94.2 \pm 0.4$ | $93.6 \pm 0.2$ | $84.1 \pm 2.1$ | $74.3 \pm 2.4$ | $86.0 \pm 0.5$ | $95.2 \pm 0.8$ | $\mathbf{95.5 \pm 0.3}$ |
|  |  | 0.5 | 8:4:2:1 | $93.8 \pm 0.4$ | $93.4 \pm 0.2$ | $92.9 \pm 0.4$ | $73.6 \pm 1.9$ | $93.4 \pm 0.2$ | $94.2 \pm 1.0$ | $\mathbf{96.8 \pm 0.6}$ |
|  |  |  | 4:2:1:1 | $91.2 \pm 0.4$ | $91.4 \pm 0.3$ | $91.1 \pm 1.6$ | $73.1 \pm 1.6$ | $91.6 \pm 0.4$ | $\mathbf{93.8 \pm 0.9}$ | $93.8 \pm 0.5$ |
|  |  | 0.2 | 8:4:2:1 | $90.2 \pm 0.3$ | $92.7 \pm 0.2$ | $92.8 \pm 0.4$ | $80.1 \pm 2.7$ | $93.8 \pm 0.1$ | $95.3 \pm 1.8$ | $\mathbf{96.2 \pm 0.5}$ |
|  |  |  | 4:2:1:1 | $92.1 \pm 0.7$ | $93.4 \pm 0.2$ | $94.8 \pm 0.6$ | $82.7 \pm 1.9$ | $94.0 \pm 0.1$ | $\mathbf{98.0 \pm 0.1}$ | $97.1 \pm 0.3$ |
| STL–10 | STL-10-1 | 0.8 | 9:1 | $73.6 \pm 1.7$ | $62.9 \pm 0.3$ | $74.4 \pm 1.6$ | $74.5 \pm 1.2$ | $73.2 \pm 1.4$ | $79.6 \pm 0.1$ | $\mathbf{80.9 \pm 0.6}$ |
|  |  |  | 1:1 | $83.0 \pm 0.9$ | $70.0 \pm 0.1$ | $85.4 \pm 0.6$ | $79.2 \pm 2.5$ | $80.0 \pm 0.7$ | $\mathbf{88.9 \pm 0.6}$ | $88.1 \pm 0.4$ |
|  |  |  | 1:9 | $82.7 \pm 1.1$ | $67.0 \pm 0.3$ | $83.8 \pm 1.4$ | $74.1 \pm 1.3$ | $78.1 \pm 1.0$ | $84.1 \pm 1.3$ | $\mathbf{87.9 \pm 0.4}$ |
|  |  | 0.5 | 9:1 | $80.4 \pm 2.4$ | $82.1 \pm 1.0$ | $83.4 \pm 2.8$ | $74.9 \pm 2.5$ | $83.7 \pm 1.7$ | $86.3 \pm 0.8$ | $\mathbf{91.3 \pm 0.8}$ |
|  |  |  | 1:1 | $87.1 \pm 1.4$ | $88.1 \pm 0.2$ | $91.9 \pm 0.7$ | $87.0 \pm 0.9$ | $91.6 \pm 0.9$ | $\mathbf{96.1 \pm 0.8}$ | $93.9 \pm 0.9$ |
|  |  |  | 1:9 | $85.8 \pm 2.1$ | $86.8 \pm 1.1$ | $90.6 \pm 1.0$ | $76.8 \pm 0.8$ | $90.4 \pm 1.6$ | $88.9 \pm 0.7$ | $\mathbf{92.6 \pm 0.2}$ |
|  |  | 0.2 | 9:1 | $84.0 \pm 1.6$ | $85.4 \pm 1.1$ | $88.9 \pm 0.4$ | $83.9 \pm 1.7$ | $88.0 \pm 0.8$ | $89.1 \pm 0.6$ | $\mathbf{91.4 \pm 0.2}$ |
|  |  |  | 1:1 | $86.1 \pm 0.8$ | $88.9 \pm 1.0$ | $92.4 \pm 0.7$ | $90.0 \pm 1.1$ | $91.6 \pm 0.4$ | $\mathbf{96.8 \pm 0.8}$ | $96.5 \pm 0.1$ |
|  |  |  | 1:9 | $79.4 \pm 1.5$ | $86.2 \pm 1.2$ | $91.3 \pm 0.2$ | $82.8 \pm 0.6$ | $90.1 \pm 0.5$ | $93.1 \pm 0.8$ | $\mathbf{95.4 \pm 0.4}$ |
|  | STL-10-2 | 0.8 | 9:1 | $71.0 \pm 1.3$ | $73.1 \pm 1.8$ | $75.9 \pm 1.8$ | $79.5 \pm 1.6$ | $82.6 \pm 1.2$ | $82.3 \pm 0.9$ | $\mathbf{83.6 \pm 0.2}$ |
|  |  |  | 1:1 | $80.5 \pm 1.1$ | $83.3 \pm 0.8$ | $84.9 \pm 1.7$ | $84.6 \pm 0.4$ | $86.0 \pm 1.8$ | $\mathbf{89.9 \pm 0.6}$ | $88.5 \pm 0.6$ |
|  |  |  | 1:9 | $76.0 \pm 1.5$ | $77.6 \pm 0.6$ | $87.3 \pm 1.2$ | $82.5 \pm 1.9$ | $84.0 \pm 1.8$ | $86.5 \pm 0.3$ | $\mathbf{92.3 \pm 0.9}$ |
|  |  | 0.5 | 9:1 | $84.4 \pm 2.0$ | $90.9 \pm 1.1$ | $92.9 \pm 1.5$ | $79.7 \pm 1.8$ | $89.5 \pm 1.7$ | $93.4 \pm 0.5$ | $\mathbf{94.0 \pm 1.7}$ |
|  |  |  | 1:1 | $91.0 \pm 1.5$ | $93.1 \pm 0.5$ | $94.1 \pm 1.7$ | $85.9 \pm 1.6$ | $92.0 \pm 0.3$ | $\mathbf{96.2 \pm 0.8}$ | $95.0 \pm 0.2$ |
|  |  |  | 1:9 | $88.2 \pm 1.3$ | $92.1 \pm 1.4$ | $93.3 \pm 0.7$ | $85.2 \pm 1.1$ | $90.7 \pm 1.3$ | $94.5 \pm 0.3$ | $\mathbf{95.4 \pm 0.2}$ |
|  |  | 0.2 | 9:1 | $78.0 \pm 1.3$ | $80.0 \pm 0.4$ | $84.2 \pm 0.9$ | $86.8 \pm 1.6$ | $93.7 \pm 1.8$ | $93.1 \pm 1.3$ | $\mathbf{96.1 \pm 0.6}$ |
|  |  |  | 1:1 | $85.8 \pm 0.6$ | $88.6 \pm 0.4$ | $95.5 \pm 0.4$ | $92.4 \pm 1.0$ | $95.7 \pm 1.0$ | $96.6 \pm 0.6$ | $\mathbf{96.8 \pm 0.6}$ |
|  |  |  | 1:9 | $84.1 \pm 0.2$ | $84.0 \pm 0.9$ | $92.7 \pm 0.2$ | $91.4 \pm 0.6$ | $92.5 \pm 1.3$ | $94.4 \pm 0.3$ | $\mathbf{95.0 \pm 0.4}$ |

Figure 3 represents the worst-case test accuracies of *PU-ScRR-CC* and other SOTA methods on Fashion–MNIST & STL–10 datasets. *PU-ScRR-CC* outperforms almost all the cases with a minimum of $1.3\%$ average improved worst-case test accuracy. The highest improvement margin is observed in fig. 3i, which is approx. $10.67\%$ on average across all considered $\alpha_{\mathsf{p}}$. Figure 3d and figure 3e indicate that *PU-ScRR* surpasses *PU-ScRR-CC* by $2.6\%$ and $2.1\%$ respectively as the top-performing model. It is noteworthy that, in both cases, the coat and shirt subclass samples are uniformly represented in $\mathcal{X}_{\mathsf{p}}$.

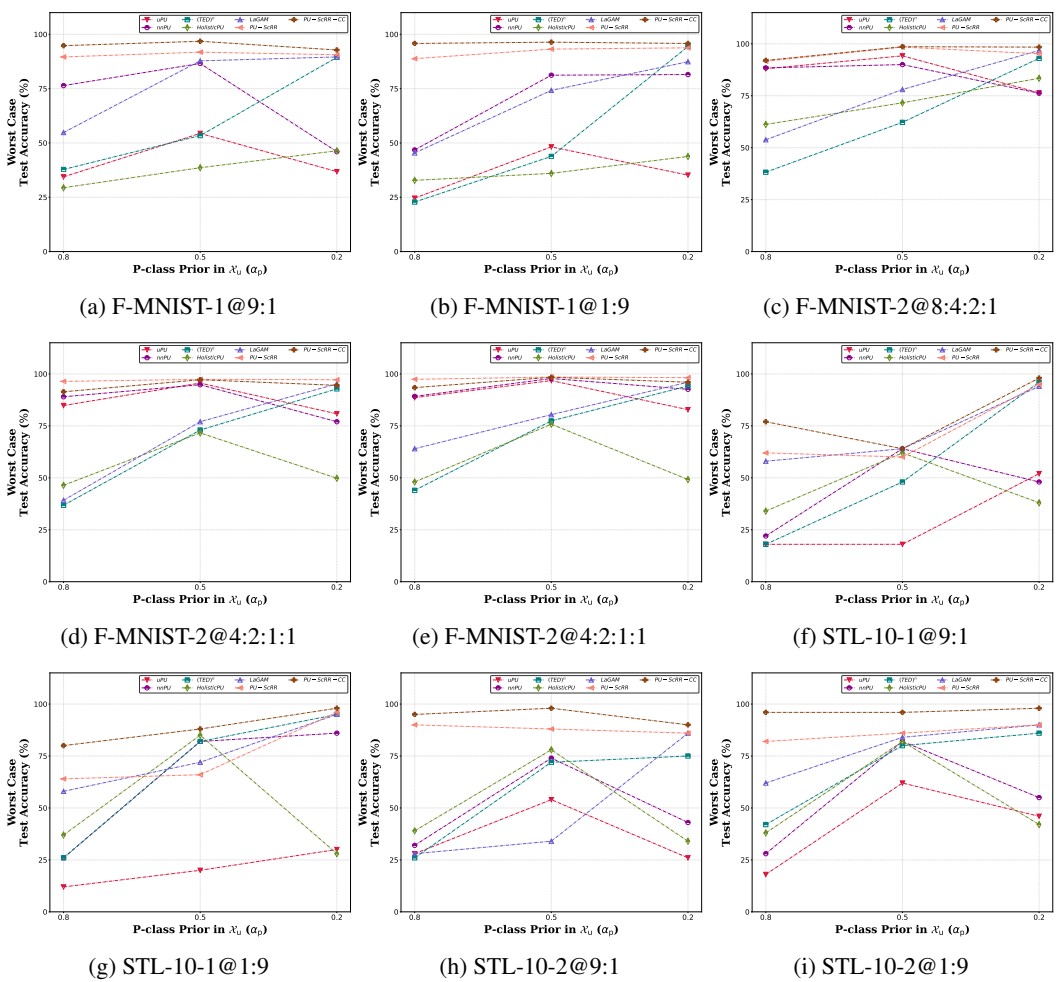

(a) F-MNIST-1@9:1      (b) F-MNIST-1@1:9      (c) F-MNIST-2@8:4:2:1

(d) F-MNIST-2@4:2:1:1      (e) F-MNIST-2@4:2:1:1      (f) STL-10-1@9:1

(g) STL-10-1@1:9      (h) STL-10-2@9:1      (i) STL-10-2@1:9

Figure 3: Comparison of worst-case test accuracies of *PU-ScRR-CC* against other SOTA methods under varying proportions of positive data in the unlabeled set $\mathcal{X}_{\mathsf{u}}$ on Fashion–MNIST and STL–10 datasets.

# 4 RELATED WORK

This section presents a review of representative studies on PU learning, with particular emphasis on those most pertinent to *PU-ScRR-CC*. Furthermore, we provide a brief discussion on the related research on the concept of hidden stratification and its applications across different domains.

## 4.1 PU LEARNING

PU learning has been actively investigated for several decades due to its broad range of applications. Bekker and Davis (Bekker & Davis, 2020) provided a comprehensive survey summarizing the key developments and practical use cases of PU learning.

Early research predominantly adopted the ***two-step approach*** for solving PU learning problems, particularly in text classification. This strategy first identifies reliable negative samples from the unlabeled set and subsequently uses them in a supervised learning setup. Representative methods following this paradigm include (Li & Liu, 2003; Yu et al., 2004; Shunxiang et al., 2023). However, the performance of such methods is highly sensitive to misidentified negatives, which can severely degrade classifier accuracy.

Recent approaches formulate PU learning by assigning weights to unlabeled instances, which represent the probability of belonging to the positive or negative class (Lee & Liu, 2003; Liu et al., 2005; Zhang & Lee, 2005). Accurate estimation of the class-prior probability $\alpha_p$ is critical for reliable weighting, yet its empirical estimation is error-prone, often degrading performance. To address these limitations, recent works have introduced *unbiased risk estimators* (Du Plessis et al., 2015; Kiryo et al., 2017; Garg et al., 2021), which bypass manual weight tuning and provide improved results. Later, abs-PU (Hammoudeh & Lowd, 2020) replaced the max-term with an absolute value penalty, simplifying optimization while achieving comparable or slightly superior accuracy. $(TED)^n$ (Garg et al., 2021) integrates class-prior estimation (BBE) with a simple CVIR objective in an iterative manner, showing consistent improvements across benchmarks. Other recent advances include *HolisticPU* (Xinrui et al., 2023), which resamples positive data and tracks predictive trends to refine labels, *LaGAM* (Long et al., 2024), which uses hierarchical contrastive learning and meta-learning for robust label refinement.

### 4.2 HIDDEN STRATIFICATION

The phenomenon of ***hidden stratification*** arises when coarse class labels conceal semantically meaningful subclasses that exhibit highly variable performance. Oakden-Rayner et al. (Oakden-Rayner et al., 2020) first demonstrated that models achieving high overall accuracy may still underperform on rare yet clinically critical subclasses. They proposed three complementary strategies to *measure* hidden stratification: *schema completion*, which exhaustively labels fine-grained subclasses on the test set, *error auditing*, which manually inspects systematic failure patterns, and *algorithmic discovery*, which applies unsupervised clustering in the learned feature space. These methods revealed substantial subclass-level performance gaps in medical imaging and vision benchmarks. While schema completion and auditing offer precise assessments, they require costly expert annotation, whereas clustering can miss subclasses that are not well separated in feature representations. This work underscored the need for principled mitigation methods beyond mere diagnosis.

Several methods address underrepresented subpopulations. SBL (Chen et al., 2019) preserves accuracy on user-defined slices but assumes known subgroups. SKD (Sajedi et al., 2022) distils subclass knowledge from teacher to student models but needs subclass labels. PromptAttack (Metzen et al., 2023) synthesizes rare subgroup examples using text-to-image generation but is prompt-sensitive and computationally expensive.

Hidden stratification is crucial in medical applications. Zeng et al. (Zeng et al., 2023) study lung nodule malignancy classification using spiculation-, clustering-, and malignancy-based stratification. Poles et al. (Poles et al., 2024) propose a Convolutional AutoEncoder $K$-means approach for osteoporosis grading, effective but domain-specific and less generalizable to other modalities

## 5 CONCLUSION

This work introduced the subclass-aware PU learning method *PU-ScRR-CC*, designed to address hidden stratification in positive–unlabeled classification. By explicitly modeling latent subclass structures within the labeled positive data, the approach delivers superior performance compared to state-of-the-art PU methods—not only in terms of overall accuracy but also in terms of worst-case accuracy with respect to rare positive subclasses. *PU-ScRR-CC* avoids the need for any hyperparameter tuning and is computationally more efficient, making it attractive in large-scale applications.

The method is particularly effective under skewed subclass distributions, where rare subclasses are underrepresented in the labeled positive data, yet it also maintains competitive performance when subclass prevalence is uniform. The results demonstrate that incorporating subclass-aware potential negative sample extraction significantly improves the robustness and reliability of PU classifiers across diverse real-world scenarios.

AUTHOR CONTRIBUTIONS

ACKNOWLEDGMENTS

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

# A APPENDIX

## A.1 DETAILS OF THE STATE-OF-THE-ART (SOTA) METHODS

We have performed experiments to compare *PU-ScRR-CC* with 5 SOTA and open-source PU learning approaches. The compared PU learning approaches include *uPU* (Du Plessis et al., 2015), *nnPU* (Kiryo et al., 2017), $(TED)^n$ (Garg et al., 2021), *HolisticPU* (Xinrui et al., 2023), and *LaGAM* (Long et al., 2024). A brief description of those compared PU learning algorithms is provided here.

Du Plessis et al. introduced an unbiased risk estimator based on non-convex loss functions satisfying a specific symmetry condition, which requires the mixture proportion $\alpha_p$, and later Du Plessis et al. extended this approach to convex loss functions, which is known as **uPU** in PU learning literature. Recognizing the risk of overfitting in modern overparameterized models, Kiryo et al. proposed **nnPU** a regularized approach that suppresses the loss on unlabeled data by clipping it at zero.

$(TED)^n$ is a combination of two methods: (i) *BBE*—used to determine the ratio of positive examples in the unlabeled data; (ii) *CVIR*-a simple and efficient objective for PU-learning. $(TED)^n$ combines these two methods in an iterative manner and outperforms across various benchmarks.

**HolisticPU** monitors prediction dynamics during training to compute a predictive trend score. This score is used to resample positive data and infer labels for unlabeled examples, addressing class

imbalance and noisy supervision. Extensive experiments show improved accuracy and robustness over existing PU learning methods, narrowing the gap to fully supervised learning.

***LaGAM*** framework uses hierarchical contrastive learning to extract latent group semantics for better feature representation. It applies meta-learning–based iterative label refinement to reduce label noise and improve robustness in PU learning.

The source codes of these 5 SOTA PU learning algorithms *i.e.*, *uPU*[1], *nnPU*[1], *(TED)$^n$*[2], *HolisticPU*[3], and *LaGAM*[4] are publicly available online.

### A.2 EXPERIMENTAL SETUP

In this section, we discuss the following things:

1. the strategy used to construct the labeled positive set $\mathcal{X}_{\mathsf{p}}$ and the unlabeled set $\mathcal{X}_{\mathsf{u}}$ for each classification task, and

2. the evaluation metrics that are used to assess the PU learning algorithms

#### CONSTRUCTION OF $\mathcal{X}_{\mathsf{p}}$ AND $\mathcal{X}_{\mathsf{u}}$

The training dataset consists of a labeled positive set $\mathcal{X}_{\mathsf{p}}$, containing $|\mathcal{X}_{\mathsf{p}}|$ labeled positive samples, and an unlabeled set $\mathcal{X}_{\mathsf{u}}$, comprising $|\mathcal{X}_{\mathsf{u}}|$ samples drawn from both positive and negative classes. In the subsequent analysis, the class labels within $\mathcal{X}_{\mathsf{u}}$ are intentionally discarded. The labeled positive samples, denoted by $\{x_i^{\mathsf{p}}\}_{i=1}^{|\mathcal{X}_{\mathsf{p}}|}$, are randomly selected from the overall population of positive examples. The *positive class prior* in $\mathcal{X}_{\mathsf{u}}$, denoted by $\alpha_{\mathsf{p}}$, represents the proportion of positive samples within the unlabeled set. To construct $\mathcal{X}_{\mathsf{u}}$, a random selection of $\alpha_{\mathsf{p}} \cdot |\mathcal{X}_{\mathsf{u}}|$ positive samples and $(1 - \alpha_{\mathsf{p}}) \cdot |\mathcal{X}_{\mathsf{u}}|$ negative samples is drawn from their respective populations.

The sizes of the training set ($\mathcal{X}^* = \mathcal{X}_{\mathsf{p}} \cup \mathcal{X}_{\mathsf{u}}$) and the test set for all considered experiments are summarized in Table 3. To better reflect practical scenarios, we evaluate performance using three different values of $\alpha_{\mathsf{p}}$, defined as $\mathcal{D}_{\alpha_{\mathsf{p}}} = \{0.8, 0.5, 0.2\}$. These values are chosen to analyze how the classifier behaves when $\mathcal{X}_{\mathsf{u}}$ is dominated by positive samples ($\alpha_{\mathsf{p}} = 0.8$), balanced ($\alpha_{\mathsf{p}} = 0.5$), or dominated by negative samples ($\alpha_{\mathsf{p}} = 0.2$).

The objective also requires how the classifier's performance is affected by the different distributional presence of the hidden subclasses in $\mathcal{X}_{\mathsf{p}}$. Hence, we have considered 3 different class label ratios for all the experiments where only 2 hidden subclasses $(z_1, z_2)$ are present in $\mathcal{X}_{\mathsf{p}}$.

1. **Ratio 1** (9: 1): Dominating subclass is $z_1$ and $z_2$ is the rare subclass.

2. **Ratio 2** (1: 1): Both $z_1$ and $z_2$ are uniformly present.

3. **Ratio 3** (1: 9): Dominating subclass is $z_2$ and $z_1$ is the rare subclass.

Classification tasks as CIFAR-10-3, CIFAR-100-2, and F-MNIST-2 involve 4, 5, 4 subclasses respectively in $\mathcal{X}_{\mathsf{p}}$. The considered class label ratios for CIFAR-10-3 and F-MNIST-2 are $8\colon 4\colon 2\colon 1$ and $4\colon 2\colon 1\colon 1$. Similarly, the class label ratios for CIFAR-100-2 are considered as $5\colon 4\colon 3\colon 2\colon 1$ and $1\colon 1\colon 1\colon 1\colon 1$. These varying subclass ratios are chosen to evaluate the classifier's performance under scenarios where certain subclasses dominate, others are rare, and all subclasses are equally represented.

#### METRIC USED FOR EVALUATION

To assess the performance of *PU-ScRR-CC* and other SOTA methods, we use test accuracy considering all subclasses as the evaluation metric. This metric is termed as **overall test accuracy**, which is standard for binary classification tasks. The reported results correspond to the mean overall test accuracy over four random seeds, along with the standard deviation.

---

[1] https://github.com/kiryor/nnPUlearning
[2] https://github.com/acmi-lab/PU_learning
[3] https://github.com/wxr99/HolisticPU
[4] https://github.com/llong-cs/LaGAM

We introduce another evaluation metric, termed **worst-case test accuracy**, which measures the classifier's accuracy on the rarest subclass. Assessing performance on such underrepresented subclasses is a key focus of this work, as worst-case test accuracy provides a more reliable indication of the model's behaviour on rare but critical cases than the average overall test accuracy across the dataset.

## MODEL ARCHITECTURE

In PU learning, various classifiers effectively handle benchmark datasets. Our method uses a hybrid model based on feed-forward neural networks (FFNN) with ReLU activations. We employ a pre-trained ResNet18 (He et al., 2016) for all datasets. A 5-layer FFNN $\varsigma(\cdot)$ with architecture $d$-1024-512-1024-256-$(k+1)$, where $d$ is the feature dimension from the base model, is appended during the *Warm-Start* phase. The $k$-value for *PU-ScRR* is fixed at 1. In *PU-ScRR-CC*, $k$ corresponds to the number of connected components in $\mathcal{X}_\mathsf{p}$, where each component represents a group of strong neighbouring samples under a chosen distance metric. After extracting potential negatives, both labeled positive and potential negative samples are reprocessed through the same base model and passed to two FFNNs: $\omega_1(\cdot)$ (same as $\varsigma$) and $\omega_2(\cdot)$. To ensure fairness, all PU algorithms use features from the same pre-trained backbone.

## CONFIGURATION

All experiments are implemented in PyTorch (Paszke et al., 2019) and executed on an NVIDIA A100 GPU. *PU-ScRR-CC* require a warm start to identify potential negative samples and are pretrained for 5 epochs in each experiment. According to (Xinrui et al., 2023), *HolisticPU* uses 15 warm-up epochs to estimate unlabeled sample trend scores for resampling, while (Long et al., 2024) reports that *LaGAM* uses 20 warm-up epochs to stabilize representation learning before applying meta-learning. Stochastic Gradient Descent with a 0.9 momentum is used as the optimizer for all experiments on CIFAR-10, CIFAR-100, Fashion–MNIST, and STL-10. The learning rate is selected via grid search in [0.001, 0.1], and the hyperparameter $\gamma$ in $\widehat{R}_{\mathsf{PU-ScRR}}$ is tuned independently over [0.001, 1]. For CIFAR-10 and Fashion–MNIST, training runs for 50 epochs per seed with a batch size of 32; CIFAR-100 and STL-10 use the same setup with a batch size of 16. We report mean overall test accuracy with standard deviation as well as the worst-case test accuracy across all seeds.

## CLASSIFICATION TASKS

The detailed specifications of the conducted experiments are listed in Table 3.

| Dataset | Task Name | Pos. Class | Neg. Class | Train Set | | Test Set |
|---|---|---|---|---|---|---|
| | | | | $\|\mathcal{X}_\mathsf{p}\|$ | $\|\mathcal{X}_\mathsf{u}\|$ | # samples |
| CIFAR-10 | CIFAR-10-1 | Bird (2), Cat (3) | Airplane (0) | 1000 | 5000 | 2000 |
| | CIFAR-10-2 | Non-animal | Animal | 3750 | 15000 | 10000 |
| CIFAR-100 | CIFAR-100-1 | Dolphin (30), Seal (72) | Shark (73) | 100 | 600 | 200 |
| | CIFAR-100-2 | Aquatic mammals (0) | Fish (1) | 1000 | 1500 | 1000 |
| Fashion–MNIST | F-MNIST-1 | T-shirt/top (0), Pullover (2) | Trouser (1) | 1000 | 6000 | 2000 |
| | F-MNIST-2 | Topwear | Others | 4800 | 15000 | 5000 |
| STL-10 | STL-10-1 | Airplane (0), Car (2) | Bird (1) | 200 | 600 | 200 |
| | STL-10-2 | Bird (1), Cat (3) | Airplane (0) | 200 | 600 | 200 |

Table 3: Specification of datasets for different classification tasks

## B  DECLARATION ON THE USE OF LARGE LANGUAGE MODELS (LLMs)

The authors acknowledge the assistance of **Grammarly** for grammatical assistance to prepare the manuscript.

