# OpenReview forum: "Unmasking and Exploiting Hidden Strata for Robust and Inclusive Positive Unlabeled Learning"
_ICLR.cc/2026/Conference — ICLR 2026 Conference Withdrawn Submission_

### Official Review · Reviewer_qqZM · 2025-10-31

**Soundness:** 2
**Presentation:** 3
**Contribution:** 2
**Rating:** 4
**Confidence:** 5

**Summary:**

This paper addresses the hidden stratification problem in Positive-Unlabeled (PU) learning by proposing the PU-ScRR-CC method, which automatically discovers fine-grained subclass structures within the positive class through similarity graph-based connected component analysis and leverages this subclass information to improve potential negative selection and classifier training. Experimental results on 4 artificial datasets demonstrate improvements in both overall test accuracy and worst-case test accuracy.

**Strengths:**

(1) This paper presents a comprehensive experimental design, constructing eight binary classification tasks across four datasets, examining different class priors and various subclass distribution ratios, and introducing worst-case test accuracy as an evaluation metric with particular emphasis on the performance of rare subclasses.
(2) This paper addresses a previously neglected practical issue in PU learning, that fine-grained subclass structures within the positive class can bias classifiers toward dominant subclasses and lead to systematic misclassification of rare subclasses. The authors also claim that this work is the first to address the hidden stratification problem in the PU learning paradigm.
(3) This paper is overall well written and easy to follow.

**Weaknesses:**

(1) This paper suffers from fundamental motivational flaws. It attributes hidden stratification to the PU learning paradigm itself, when it is actually an engineering artifact resulting from artificial dataset construction rather than an inherent problem of PU learning. Moreover, while the motivation emphasizes real-world application scenarios such as medical imaging, the experiments rely entirely on artificially constructed benchmark datasets and lack validation on real hidden stratification data. This raises a critical question that does the problem definition itself have inherent issues?
(2) The core methodology essentially employs confidence-weighted loss with regularization, which represents standard techniques in semi-supervised learning, thus lacking innovation. Additionally, using 1-NN to construct graphs and extract connected components is a classical clustering approach. The paper fails to adequately justify its advantages over mature clustering methods such as spectral clustering or DBSCAN.
(3) The experimental results indicate that in numerous scenarios, PU-ScRR-CC's performance is comparable to or even inferior to PU-ScRR.
(4) This paper provides no theoretical analysis and relies solely on empirical validation to support its claims. It would be advisable to include theoretical guidance on the number of 1-NN connected components or provide formal analysis establishing the soundness of recovering semantic subclasses through this approach.

**Questions:**

N/A

---

### Official Review · Reviewer_B9Xc · 2025-11-01

**Soundness:** 2
**Presentation:** 3
**Contribution:** 2
**Rating:** 4
**Confidence:** 5

**Summary:**

This paper focuses on the positive unlabeled learning and considers the hidden stratification within real-world datasets, where the positive class comprises multiple fine-grained subclasses with varying prevalence. It proposes a subclass-aware PU learning method to discover the subclasses with a graph-based method and selects latent negative samples from unlabeled set based discovered subclasses.

**Strengths:**

1. The fine-grained subclasses within positive or/and negative classes for PU learning are important and widely existed in practice.
2. The experimental results show the effectiveness of the proposed method.
3. The paper is well-written and easy to follow.

**Weaknesses:**

1. The contribution is incremental and the novelty is limited. The fine-grained subclasses within PU learning are exploited in LaGAM [1] with group-aware representation learning based clustering and label alignment. This paper discovers the fine-grained subclasses within positive classes by clustering and selects negative samples from the unlabeled set based on subclasses. Thus, the contribution is incremental and the novelty is limited. Otherwise, when labeled positive samples are very scarce (widely exist in practice), the clustering may be very unstable and noisy. And how to deal with the noise clusters?
2. Another issue I am concerned about is how to guarantee that the standard of unsupervised clustering is consistent with one of the actual fine-grained subclasses. This will affect the label alignment among the discovered clusters and original labels.

[] Positive-Unlabeled Learning by Latent Group-Aware Meta Disambiguation. CVPR2024.

**Questions:**

Please refer to Weaknesses.

---

### Official Review · Reviewer_BMVu · 2025-11-01

**Soundness:** 2
**Presentation:** 2
**Contribution:** 2
**Rating:** 2
**Confidence:** 5

**Summary:**

This paper introduces a subclass-aware PU learning framework. The proposed method first identifies hidden subclasses via a graph-based clustering technique, and then systematically leverages these discovered subclasses to extract high-confidence negative candidates from the unlabeled set. Experimental results across diverse datasets suggest that this approach can lead to performance benefits compared to listed PU learning methods under varying subclass distributions.

**Strengths:**

- The paper is well-organized and clearly written, making the problem setup and methodology easy to follow.
- It introduces the concept of "hidden subclass" into the PU learning and provides a framework that discovers and leverages latent subclasses within the positive set.

**Weaknesses:**

- The proposed method is critically dependent on the quality of the pre-trained feature extractor for its initial clustering stage.
- While the "hidden subclasses" in the datasets are pre-defined with fine-grained labels, their definitions can be far more ambiguous in real-world data. The method defines subclasses as connected components, which may correspond to either semantic subclasses or merely incidental clusters in the feature space, with no mechanism to distinguish between these two fundamentally different cases.
- The method was not tested on standard datasets in their canonical setup. Furthermore, the comparisons lack benchmarks against several established classical PU learning methods:

[1] Positive-unlabeled learning with label distribution alignment (TPAMI2023)

[2] Positive-unlabeled learning from imbalanced data (IJCAI2021)

[3] A variational approach for learning from positive and unlabeled data (NeurIPS2020)

**Questions:**

Please refer to the weaknesses.

---

### Note · Authors · 2025-11-18

I have read and agree with the venue's withdrawal policy on behalf of myself and my co-authors.